# Napyradiomycin A4 and Its Relate Compounds, a New Anti-PRV Agent and Their Antibacterial Activities, from *Streptomyces kebangsaanensis* WS-68302

**DOI:** 10.3390/molecules28020640

**Published:** 2023-01-08

**Authors:** Yani Zhang, Wei Fang, Kaimei Wang, Zhigang Zhang, Zhaoyuan Wu, Liqiao Shi, Fang Liu, Zhongyi Wan, Manli Liu

**Affiliations:** Hubei Biopesticide Engineering Research Centre, Hubei Academy of Agricultural Sciences, No. 8, Nanhu Ave., Hongshan District, Wuhan 430064, China

**Keywords:** napyradiomycins, anti-PRV, antibacterial, *Streptomyces kebangsaanensis* WS-68302

## Abstract

Two new napyradiomycins derivatives, napyradiomycin A4 (**1**) and A80915 H (**2**), along with five known ones, were isolated from the ethyl acetate extract of fermentation culture of *Streptomyces kebangsaanensis* WS-68302. Their structures were elucidated by extensive spectroscopic analysis, including HR-MS, 1D and 2D NMR, CD spectrum, as well as comparison with literature data. Compound **1** exhibited significant antiviral activity against PRV (Pseudorabies virus) with an IC_50_ value of 2.056 μM and therapeutic ratio at 14.98, suggesting that it might have potential for development of an antiviral agent. Moreover, compound **1** displayed the strongest inhibition against PRV protein among the tested napyradiomycins in the indirect immunofuorescence assay. Compounds **3** and **4** showed higher activities against swine pathogenic *Streptococcus suis* than the positive control penicillin G sodium salt, with MIC values of 3.125 and 6.25 μg/mL, respectively. Compounds **1** and **3**–**6** exhibited moderate antibacterial activity against the swine pathogenic *Erysipelothrix rhusiopathiae*, with MIC values ranging from 25 to 50 μg/mL.

## 1. Introduction

In recent decades, infection with pathogens has caused huge economic losses to the swine industry. Pseudorabies virus (PRV), a herpes virus, is one of the most important pathogens in swine [1,2]. The swine pseudorabies caused by the pseudorabies virus is characterized by reproductive disturbance, dyspnea, and neurological symptoms, and different-aged pigs are susceptible to infection with the disease [3,4]. Vaccination is one of the most important methods to prevent and control PRV; however, with the emergence of PRV mutant strains in recent years, it is difficult to completely prevent and control them using conventional vaccines [5,6,7,8]. Moreover, no effective anti-PRV drugs have been approved [9]. Therefore, it is essential to discover antiviral drugs. 

In addition to the virus, some bacteria, such as *Streptococcus suis* and *Erysipelothrix rhusiopathiae*, can also cause economic losses and zoonotic diseases. *S. suis* is an important zoonotic pathogen that causes several pathological conditions such as septicemia, meningitis, arthritis and pneumonia [10]. *S. suis* can cause bacterial encephalitis or toxic-shock-like syndrome in humans [11]. *E. rhusiopathiae*, a small Gram-positive bacterium, is also a zoonotic pathogen [12]. The swine erysipelas caused by *E*. *rhusiopathiae* mainly has acute, subacute and chronic clinical forms, and acute infection can even cause sudden death or general signs of septicemia to swine [13]. *E. rhusiopathiae* can also cause a skin disease, erysipeloid, in humans [13]. Although many antibiotic drugs can treat bacterial diseases, with the emergence bacterial resistance, the development of new drugs to prevent and treat these diseases is urgent [14]. 

Napyradiomycins are a large class of unique meroterpenoids including a semi-naphthoquinone nucleus and an isoprene unit or a monoterpenoid subunit. Napyradiomycins initially attracted attention owing to their strong antimicrobial properties toward Gram-positive bacteria [15,16,17,18,19]. Subsequently, napyradiomycins were reported to exhibit a wide array of biological activities, including gastric (H+-K+) ATPase-inhibiting activities [20], estrogen receptor antagonist activities [21], apoptosis-inducing activities [22], and cytotoxicity [23,24]. In recent years, the napyradiomycins produced by marine-derived actinomycetes have been found to have good antifouling activities [25]. In addition, a new regioisomer of napyradiomycin A1, 16*Z*-19-hydroxynapyradiomycin A1, has been confirmed to possess antioxidant and anti-inflammatory properties [26]. However, there have been no reports on inhibitory virus activities of napyradiomycins. 

In our ongoing research on the bioactive secondary metabolites from actinomycetes, a soil-derived strain WS-68302 was found to show interesting antiviral activity against PRV and antibacterial activity. The actinomycete strain was identified as *S. kebangsaanensis* WS-68302. Further chemical investigation of the extract from the strain led to the discovery of two new derivatives of napyradiomycins, named napyradiomycin A4 (**1**) and A80915H (**2**), as well as five known ones (**3**–**7**) (Figure 1). The napyradiomycins (**1** and **3–6**) showed strong anti-PRV activity in vitro. In this study, we report the taxonomy of the strain, the isolation, the structural elucidation, and the antiviral and antibacterial activities of these compounds from the strain. 

## 2. Results and Discussion

### 2.1. Taxonomy of Strain WS-68302

The colonies of strain WS-68302 on ISP-2 agar grew abundant grayish white mycelium (Figure 2A). The aerial mycelia were straight to flexuous and spiral on terminals. (Figure 2B). The spores were mostly ovoid and 0.8–1.2 × 1.2–2.0 μm in size, and the surfaces of the spores were covered with different-length spines (Figure 2C). These morphological characteristics of strain WS-68302 were in total agreement with the strain *Streptomyces kebangsaanensis SUK12*, which was identified as a novel species of the genus Streptomyces in 2013, isolated from Malaysian ethnomedicinal plant *Portulaca olerace* [27]. The 16S rRNA sequence of the strain WS-68302 was deposited in the NCBI GenBank database (accession number OP787989), and the BLAST result of the sequence also verified the closest similarity (98.99%) to that of *S. kebangsaanensis* SUK12. The phylogenetic tree generated using the neighbor-joining method clearly depicted the evolutionary relationship of the strain WS-68302 with *Streptomyces* (Figure 3). Accordingly, the strain WS-68302 was identified as *S. kebangsaanensis* and designated *S*. *kebangsaanensis* WS-68302.

### 2.2. Structural Elucidation

Compound **1** was obtained as pale yellow oil. In the electrospray ionization mass spectrometry spectrum (ESIMS), compound **1** showed a positive ion cluster at *m*/*z* 497.4/499.2/501.4 in the ratio of 100:66:10, indicating the presence of two chlorine atoms. HRESIMS analysis gave an ion consistent with the molecular formula C_25_H_30_Cl_2_O_6_Na (calculated for C_25_H_31_Cl_2_O_6_Na 519.1312 [M + Na]^+^). The IR spectrum of **1** displayed bands at 3360 cm^−1^ for hydroxyl groups, 1697 cm^−1^ for conjugated carbonyl groups. The ^1^H NMR spectrum (Table 1) of **1** revealed an exchangeable proton singlet at *δ*_H_ 11.85, two broad singlet aromatic protons at *δ*_H_ 7.17 and 6.72, a triplet of methine doublet proton at *δ*_H_ 4.70 (d, *J* = 8.5 Hz), a double coupled methine proton [*δ*_H_ 4.42 (dd, *J* = 12.0, 4.0 Hz)], a methine multiple proton at *δ*_H_ 2.53 (m), four sets of methylene protons [*δ*_H_ 2.68 (brd, *J* = 8.1 Hz), 2.48 (dd, *J* = 14.3, 4.0 Hz) and 2.43 (d, *J* = 12.0 Hz), 2.15 (m) and 2.10 (m), and 1.93 (m) and 1.88 (m)], and five methyl protons [*δ*_H_ 1.50 (s), 1.32 (s), 1.18 (s), 1.06 (t, *J* = 6.4 Hz), and 1.06 (t, *J* = 6.4 Hz)]. The ^13^C NMR spectrum (Table 1) of **1** showed resonances for three carbonyl carbons (*δ*_C_ 215.0, 195.5 and 194.0), two oxygenated aromatic carbons (*δ*_C_ 164.8 and 164.4), three unsaturated methine carbons (*δ*_C_ 115.7, 109.5, and 108.2), six quaternary carbons (*δ*_C_ 141.3, 135.4, 109.9, 83.7, 79.2, and 78.9), two aliphatic methine carbons (*δ*_C_ 58.9 and 41.1), four methylene carbons (*δ*_C_ 42.8, 41.2, 38.0, and 33.3), and five methyl carbons (*δ*_C_ 29.0, 22.4, 18.4, 18.3, and 16.7). Combined with the 2D-NMR spectrum of **1** (Appendix A), these NMR data resembled those of napyradiomycin A1 (**3**) [16], except for the presence of a keto carbonyl carbon at C-16 (*δ*_C_ 215.0) and an aliphatic methine group at C-17 [(*δ*_C_ 41.1, *δ*_H_ 2.53 (1H, m)] in **1,** and the absence of the signal of an olefinic proton at *δ*_H_ 4.89 (brs, H-16) and the terminal double bond at C-16 (*δ*_C_ 123.9) and C-17 (*δ*_C_ 131.9) in **3**. The orthotropic ^1^H-^1^H COSY correlations (Figure 4) from 17-CH_3_ to H-17 and HMBC correlations (Figure 4) from H-17, 17-CH_3_, H-14 and H-15 to C-16, as well as H-17 to 17-CH_3_, support the above conclusion. Thus, the planar structure of **1** could be designated as napyradiomycin A4 (Figure 4). The coupling constant of H-3 (dd, *J* = 12.0, 4.0 Hz) indicated that orientation of H-3 was axial. The correlations between H-11 and 2-CH_3_ ax (*δ* 1.18), H-11 and H-4 ax (*δ* 2.43) (Appendix A) suggested that the relative configuration of C-3 and C-10a was syn. Compound **1** showed that characteristic Cotton effects (CEs) (Appendix A) fit well with those of napyradiomycin A1 (**3**) (Appendix A), indicating that the absolute configurations of benzoquinone of **1** were the same as those of napyradiomycin A1. Therefore, the absolute configurations of **1** were determined as 3*R*, 4a*R*, and 10a*S*.

Compound **2** was obtained as yellow oil. Its molecular formula was determined to be C_25_H_30_O_7_ on the basis of ^13^C NMR and HRESIMS (*m*/*z* 443.2067 [M + H]^+^, calcd for C_25_H_31_O_7_ 443.2064 [M + H]^+^). Comparison of the NMR spectroscopic data (Table 2) of **2** with those of A80915G (**5**) revealed that a methene group in A90915G (**5**) is replaced by the hydroxymethine group [*δ*_C_ 77.0; *δ*_H_ 4.12 (1H, t, *J* = 6.7 Hz)] in **2,** and a methyl group in A90915G (**5**) is replaced by the hydroxymethyl group [*δ*_C_ 68.9; (*δ*_H_ 3.97 (2H, m)] in **2**. The ^1^H-^1^H COSY correlations (Figure 4) of H-4″/H-5″/H-6″, and the HMBC correlations (Figure 4) from H-4″ to C-6″, H-5″ to C-4″ and C-6″, as well as H-10″ to C-3″ and C-4″, confirmed the hydroxy-bearing methine at C-4″. The hydroxymethyl group was assigned to C-8″ according to the HMBC correlations from H_2_-8″ to C-6″, C-7″, and C-9″. Thus, the structure of compound **2** is depicted in Figure 4, and named A80915H.

The absolute configurations at C-2 and C-3 of 2 were determined to be 2*S* and 3*S* by comparing its CD spectrum (Appendix A) with that of A80915G (Appendix A), whose structure was confirmed by spectroscopic data and chemical synthesis [19,28]. However, the absolute configuration of C-4″ was not determined due to scarcity of **2** for modified Mosher’s analysis [29].

Compounds **3**–**7** were identified as napyradiomycin A1 (**3**) [16], napyradiomycin B1 (**4**) [16], A80915G (**5**), A80915G-8″-acid (**6**) [30], and naphthomevalin (**7**) [31,32] (Figure 1), respectively, by comparing their MS, ^1^H and ^13^C ΜMR spectroscopic data with those previously reported.

### 2.3. Biological Activities

#### 2.3.1. Antiviral Activities of 1–7 against PRV In Vitro

The in vitro antiviral activities of **1**–**7** against PRV in Marc-145 cells were evaluated by MTT assay according to the protocol. Compounds **1** and **3**–**6** exhibited significant antiviral activity against PRV, with half maximal inhibitory concentration (IC_50_) values ranging from 2.056 to 26.47 μM, whereas **2** and **7** did not display any apparent anti-PRV activity (Table 3). It is notable that compounds **1** and **3** exhibited excellent antiviral activity against PRV, with IC_50_ values of 2.056 and 2.208 μM, respectively—much stronger than that of the positive control ribavirin (IC_50_ = 58.032 μM). However, the new compound **1** showed a higher therapeutic ratio (TI = 14.98) than **3** (TI = 2.62). Although their structures are closely similar, there is a carbonyl group at C-16 for compound **1** and a double bond at C-16 and C-17 for compound **2**, which led to a difference in toxicity between the two compounds and indicated that the substituent of the 16 position is important for toxicity. Despite compound **5** demonstrating a strong activity against PRV with IC_50_ value of 8.26 μM, its new derivative **2** was inactive against PRV, which suggested the hydroxyl substitution at C-4″ and C-8″ of **5** had an effect on the anti-PRV activity. Compounds **4** and **6** displayed a lower therapeutic index (TI < 2.0). As a result, compound **1** might have potential for the development of an antiviral agent.

#### 2.3.2. Indirect Immunofluorescence Assay

Based on the above antiviral activity, we selected the active compounds **1**, **3** and **4** to test the immunofluorescence effect at concentrations of 2, 5 and 10 μg/mL, respectively, by indirect immunofluorescence assay. The three compounds were co-incubated with PRV in Marc-145 cells, respectively. After 24 h, the nucleocapsid (DAPI, blue) and PRV (FITC, green) responses to the indirect immunofluorescence effect are shown in Figure 5. The green fluorescence intensity in the compound-treated cells (Figure 5B–D) was obvious lower than in the virus control group (Figure 5A), indicating that the viral protein synthesis was significantly suppressed by the three active compounds, **1**, **3** and **4**. Furthermore, the new compound **1** showed lower immunofluorescence effect against PRV and lower cytotoxicity than the known compounds **3** and **4** in the Marc-145 cells.

#### 2.3.3. Antibacterial Activities

Compounds **1**–**7** were evaluated using a microdilution broth method for antibacterial activity against *S. aureus* and the animal pathogens *S*. *suis*, *E. rhusiopathiae*, and *E. coli*. Penicillin G sodium salt and cefpirome sulfate were used as positive controls. As shown in Table 4, compounds **3**, **4** and **7** displayed potent activities comparable to the positive control cefpirome sulfate (MIC < 0.78 μg/mL) against bacteria *S. aureus* ATCC 25923, while the new compounds **1** and **6** showed moderate activity against the bacteria, with MIC values of 25 and 50 μg/mL, respectively. Compounds **3** and **4** also showed higher activities against the swine pathogenic *S*. *suis* than the positive control penicillin G sodium salt, with MIC values of 3.125 and 6.25 μg/mL, respectively, whereas the new compound **1** exhibited weak activity against the bacteria. Compounds **1** and **3**–**6** exhibited moderate antibacterial activity against the swine pathogenic *E*. *rhusiopathiae*, with MIC values ranging from 25 to 50 μg/mL. There are slight differences on a side chain in the structures of compounds **2**, **5** and **6**, but compound **2** did not show any activity against the three tested Gram-positive bacteria. Compound **5**, previously isolated from *Streptomyces aculeolatus* A80915, had good antibacterial activity against aerobic Gram-positive *Staphylococcus*, *Streptococcus* and *Enterococcus* bacterial strains (MIC 2–16 μg/mL) [19]. While compound **6**, an oxidized compound of **5**, only had weak antibacterial activity against *Bacillus subtilis* ATCC6633 (MIC 64 μg/mL) [30]. Therefore, it was suggested that hydroxyl substitution at C-4″ and C-8″of the side chains on the semi-naphthoquinone nucleus had a great influence on the activities of compound **2**. As reported in the literature [15,17,19], these tested compounds were inactive to the Gram-negative bacteria *E. coli* ATCC 25922.

## 3. Conclusions

In conclusion, seven napyradiomycins, including two new ones, napyradiomycin A4 (**1**) and A80915H (**2**), were obtained from the fermentation culture of *Streptomyces kebangsaanensis* WS-68302. To the best of our knowledge, this is the first time that the potent antiviral activities of napyradiomycins against PRV in vitro have been observed. Impressively, the new compound, napyradiomycin A4 (**1**), and napyradiomycin A1 (**3**) both showed much better inhibitory activity against PRV than ribavirin. In addition, napyradiomycin A1 (**3**) and B1 (**4**) and naphthomevalin (**7**) displayed potent activities comparable to the positive control cefpirome sulfate against the bacteria *S. aureus*. Compounds **3** and **4** also showed higher activities against swine pathogen *S. suis* than the positive control penicillin G sodium salt. This study suggested that the napyradiomycin A4 could serve as a promising antiviral agent against PRV.

## 4. Experimental Section

### 4.1. General Experimental Procedures

Optical rotations were measured in MeOH on an MPC 500 (Waltham) polarimeter (PerkinElmer Ltd., Waltham, MA, USA). UV spectrum was recorded on a Shimadzu UV-2600 PC spectrometer (Shimadzu Corporation, Kyoto, Japan). IR spectra were obtained on a NICOLET IS50 FT-IR (Thermo Fisher Scientific, Waltham, MA, USA). NMR spectra, including HSQC, HMBC and COSY, were recorded on a Bruker AC 700 MHz spectrometer using tetramethylsilane as standard (Bruker BioSpin group, Rheinstetten, German). ESIMS data was detected on a waters Xevo TQD UPLC-MS (Waters Corporation, Milford, MA, USA), with a Waters ACQUITY^®^BEH C_18_ column (2.1 × 100 mm, 1.7 μm, Ireland). HR-ESI-MS spectra: Thermo Q-T of Micromass spectrometer (Thermo Electron Corporation, Waltham, MA, USA). Preparative HPLC was carried on a Waters 2525 pump and a Waters 2998 DAD detector coupled with Waters 2767 Autopurifcation System (Waters Corporation, Milford, MA, USA) using Sunfire Prep C_18_ OBD column (19 × 250 mm, 5 μm, Waters Corporation, Milford, MA, USA).

### 4.2. Isolation of Strain and Morphological Identification

The strain WS-68302 was isolated from a soil sample collected from a hillside in SuiXian County, Hubei Province, China, in April 2015. For the taxonomic studies of the actinomycetes, ISP-2 agar (glucose 0.4%, malt extract 1%, yeast extract 0.4%, agar 1.8%, and pH 7.2) was used to investigate the morphological and physiological characteristics. The strain was incubated at 28 ℃ for 2–3 weeks. Morphological properties were observed under a scanning electron microscope (Model Hitachi SU8100). Genomic DNA extraction, PCR amplification and sequencing of 16S rRNA were developed with the universal methods [33]. The strain was identified based on sequence analysis of 16S rRNA Sequence under the BLAST search.

### 4.3. Fermentation

A stock culture of the strain *S*. *kebangsaanensis* WS-68302 was inoculated into 500 mL Erlenmeyer flasks containing 100 mL of the seed medium ISP-2. The flasks were incubated at 28 ℃ on a rotary shaker at 150 rpm, for 96 h. 10% seed culture was transferred to 500 mL Erlenmeyer flasks containing 100 mL of the production medium ISP-2 under sterile conditions. The flasks were cultivated at 28 ℃ on a rotary shaker at 150 rpm for 120 h.

### 4.4. Extraction and Isolation

The freeze-dried culture broth (30 L) *S. kebangsaanensis* WS-68302 was extracted with an equal volume of ethyl acetate 2 times. The ethyl acetate phase was combined, filtrated and evaporated under vacuum to give the crude extract (27 g). The crude extract was subjected to silica gel CC column using gradient elution with a petroleum ether and EtOAc mixture from 100/10 to 0/100 (*v*/*v*) to give ten fractions (Fr.1-Fr.10). Fr.4 (2.48 g) was purified by preparative HPLC with an OBD C_18_ column (Sunfire, 250 × 19 mm, 5 μm) to obtain **1** (3.16 mg), using a gradient solvent system (0–2 min, 5% CH_3_CN; 2–27 min, 5–100% CH_3_CN; 27–32 min, 100% CH_3_CN; 32–37 min, 100–5% CH_3_CN; 37–40 min, 5% CH_3_CN). Fr.1 (4.0 g) was further separated on silica gel CC eluting with a gradient petroleum ether–EtOAc solvent system from 20/1 to 6/1 (*v*/*v*) to afford eight fractions (Frs.1-1–Fr.1-8). Next, Frs.1-2 and Frs.1-3 were further purified by preparative HPLC to yield **3** (15.6 mg), **4** (10.87 mg), **5** (3.59 mg) and **7** (5.26 mg), under the HPLC conditions (0–2 min, 20% CH_3_CN; 2–22 min, 20–100% CH_3_CN; 22–26 min, 100% CH_3_CN; 26–30 min, 100–20% CH_3_CN; 30–33 min, 20% CH_3_CN). Fr.6 (4.4 g) was isolated and purified by preparative HPLC to obtain **6** (4.14 mg). Fr.7 (4.68 g) was subjected to silica gel CC eluting with a gradient dichloromethane and acetone solvent system from 10/1 to 1/1 (*v*/*v*) to get eleven fractions (Frs.7-1–Fr.7-11). Fr.7-7 and Fr.7-8 were also purified by preparative HPLC with an OBD C_18_ column (Sunfire, 250 × 19 mm, 5 μm) to obtain **2** (4.97 mg), with the HPC method (0–2 min, 35% CH_3_CN; 2–22 min, 35–100% CH_3_CN; 22–26 min, 100% CH_3_CN; 26–29 min, 100–35% CH_3_CN; 29–32 min, 35% CH_3_CN).

Napyradiomycin A4 (**1**). Yellow brownish oil; [α]25 D −4.2 (*c* 0.1 MeOH); IR (KBr) *υ*_max_ 3360, 2926, 2855, 2359, 1698, 1636, 1614, 1456, 1385, 1371, 1258, 1186, 1134, 1074, 1015, 862, 725 cm^−1^; UV (MeOH) λ_max_ (log *ε*): 249 (4.14), 270 (2.99), 359 (1.59) μM; ESIMS *m*/*z* 497.4, 499.2, 501.4 [M + H]^+^ (Appendix A); HRESIMS *m*/*z* 519.1316 [M + Na]^+^ (calcd. for C_25_H_30_Cl_2_O_6_Na, 519.1312) (Appendix A); ^1^H and ^13^C NMR data see Table 1. CD (*c* 0.3 MeOH): *λ*_max_ (Δ*ε*): 220 nm (−0.69), 250 nm (+0.31), 273 nm (−0.27), 300 nm (+0.23), 330 nm (−0.04), 360 nm (+0.15).

A80915H (**2**). Yellow oil; [α]25 D +59.35 (*c* 0.2 MeOH); UV (MeOH) λ_max_ (log *ε*): 251 (2.15), 270 (1.57), 363 (0.86) nm; ESIMS *m*/*z* 441.6 [M − H]^−^ (Appendix A); HRESIMS *m*/*z* 443.2067 [M + H]^+^ (calcd. for C_25_H_31_O_7_, 443.2064) (Appendix A); ^1^H and ^13^C NMR data see Table 2. CD (*c* 0.2 MeOH): *λ*_max_ (Δ*ε*): 220 nm (−0.29), 252 nm (+0.21), 270 nm (−0.03), 295 nm (+0.09), 335 nm (−0.07), 362 nm (+0.24).

### 4.5. Antiviral and Antibacterial Assays

#### 4.5.1. Antiviral Assays

##### Cells, Virus and Reagents

Marc-145 cells were grown in Dulbecco’s modified Eagle’s medium (DMEM) supplemented with 10% fetal bovine serum (FBS), 100 μg/mL streptomycin, and 100 IU/mL penicillin, at 37 °C in 5% CO_2_ atmosphere. The PRV was obtained from Professor Tan Chen of Huazhong Agricultural University. The virus was propagated in Marc-145 cells and stored at −80 ℃ until used. DMEM containing high glucose levels and Sodium Pyruvate was purchased from Hyclone. FBS was purchased from Gemini. Ribavirin, 0.25% tyrisin and MTT were purchased from Solarbio.

##### Chemicals, Antibodies and Reagents

The anti-Pseudorabies Virus antibody (ab3534), goat anti-rabbit IgG H&L (Alexa Fluor 488) (ab150077), and DAPI staining solution (ab228549) were purchased from Abcam.

##### Cell Proliferation

The supernatant was discarded upon Marc proliferation; cells were washed with PBS, and trypsin was added until the cells were digested into singles at 37 ℃. Cells were dispersed in DMEM with 10% FBS and cultured at 37 ℃ for 2–3 days.

##### Cell Bioability Assay

Cell viability was assessed using the MTT according to the protocol. Briefly, cells in 96-well plates were treated with compounds in different concentration for 48 h. Assays were performed in triplicate. Then, MTT was added to the cells and they were incubated in the dark for 4 h at 37 ℃. The supernatant was discarded and DMSO was added. After 10 min incubation, the OD 450 of each well was read using a spectrophotometer (Thermo). The cell viability (%) was calculated as (compound average OD_450μM_ value/cell control average OD_450μM_ value) × 100%. Compounds were added to monolayer in different concentration. After 48h incubation, the cell viability (%) was calculated by MTT assay, and the cytotoxic effect (CC_50_) was analyzed by SPSS 16.0 software. Compounds in different concentrations (40, 20, 10, 5, 2.5, 1.25, 0.625, 0.3125 μg/mL) and PRV (100 TCID_50_) were added to monolayer at 37℃. After 72h incubation, the cell viability (%) was calculated by MTT assay, and the half maximal inhibitory concentration (IC_50_) was analyzed by SPSS 16.0 software. Ribavirin was used as a positive control.

Therapeutic index (TI) was calculated as CC_50_/IC_50_.

##### Indirect Immunofluorescence Assay

The monolayers of Marc-145 cells cultured in 24-well plates were treated with PRV and drug mixture. After 24 h incubation, the cells were washed with PBS twice and fixed with cold absolute ethanol at 4 °C for 15 min, then permeabilized with 0.1% Triton X-100 in PBS for 10 min, and blocked with 1% BSA in PBS for 1 h. The cells were then incubated with 1:200 anti-Pseudorabies Virus antibody in PBS containing 1% BSA at 4 °C overnight. After washing three times with PBST, the cells were stained with 1:2000 goat anti-rabbit IgG H&L. To show the nucleus of the cells, DAPI (40, 6-diamidino-2-phenylindole) was used to stain the cellular DNA. The fluorescent images were observed using an inverted fluorescence microscope (Olympus IX73 (Olympus Corporation, Tokyo, Japan)).

#### 4.5.2. Antibacterial Assays

The minimum inhibitory concentrations of **1–7** were determined against the four bacterial strains (*Staphylococcus aureus* ATCC 25923, *Erysipelothrix rhusiopathiae* WH13013, *Streptococcus suis* SC19, *Escherichia coli* ATCC 25922) following the Clinical and Laboratory Standards Institute (CLSI) guidelines [34]. The microdilution broth method was performed in 96-well plates (Corning Costar^®^ 3599 Corning, Corning, NY, USA) using MHB (Hopebio, Qingdao, China). The final concentration of the culture was 5 *×* 10^5^ colony-forming units (CFU)/mL, and measurements were repeated at least in triplicate. Penicillin G sodium salt and cefpirome sulfate were purchased from Solarbio.

## Figures and Tables

**Figure 1 molecules-28-00640-f001:**
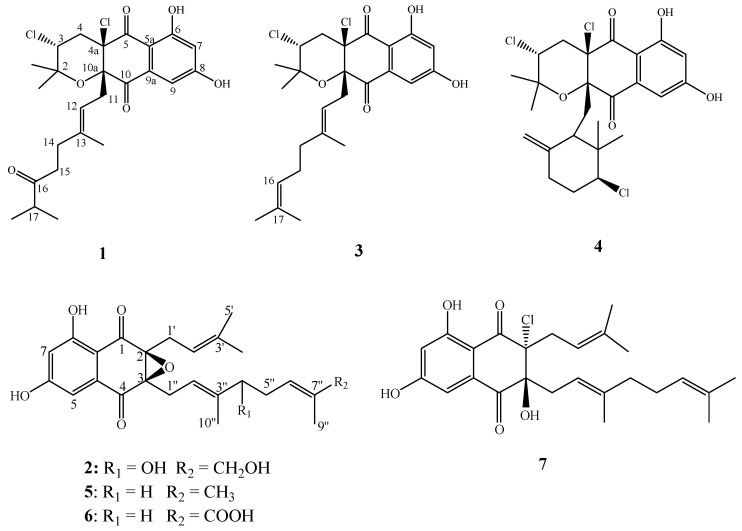
Chemical structures of compounds **1**–**7**.

**Figure 2 molecules-28-00640-f002:**
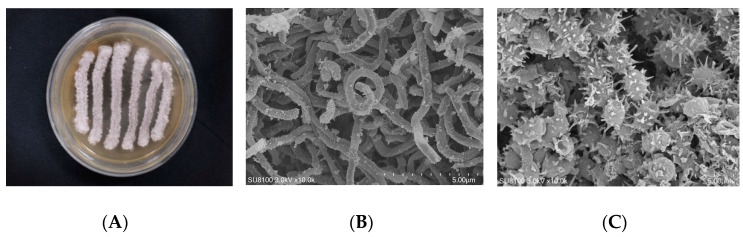
Identification of the strain WS-68302. (**A**) The colonies cultured in ISP-2 for 14 days at 28 ℃; (**B**) scanning electron micrograph of the aerial mycelium (bar, 5.0 μm); (**C**) scanning electron micrograph of the spores (bar, 5.0 μm).

**Figure 3 molecules-28-00640-f003:**
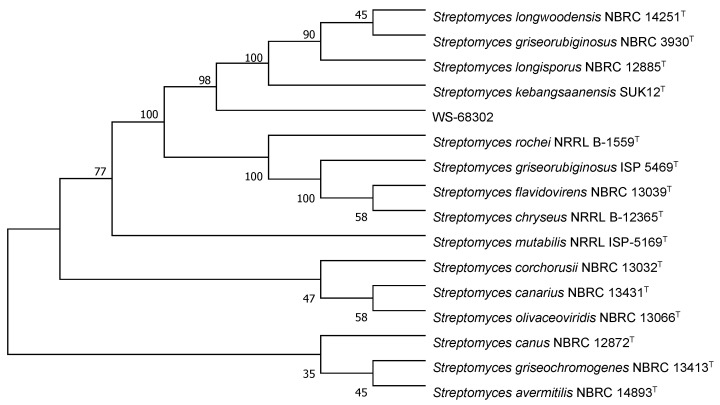
Phylogenetic tree based on 16S rRNA sequences of the *S*. *kebangsaanensis* WS-68302.

**Figure 4 molecules-28-00640-f004:**
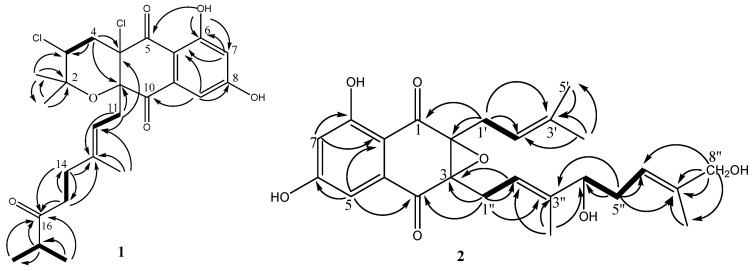
^1^H–^1^H COSY (bold lines) and HMBC (arrows) correlations of compounds **1** and **2**.

**Figure 5 molecules-28-00640-f005:**
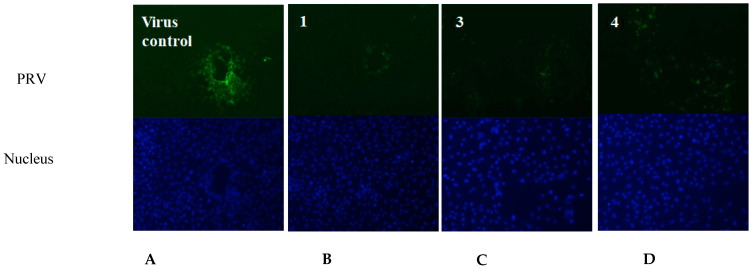
The fluorescent images of the virus control group and the compound-treated cells groups. (**A**) Virus control (PRV); (**B**) compound **1**; (**C**) compound **3**; (**D**) compound **4**. (The nucleus was stained with DAPI (blue), and the green foci indicate the presence of PRV protein).

**Table 1 molecules-28-00640-t001:** ^1^H (700 MHz) and ^13^C (175 MHz) NMR data of compound **1** in CDCl_3_.

Position	*δ* _C_	*δ*_H_ (*J* in Hz)
2	78.9, C	
2-CH_3_	29.0, CH_3_	1.50, s
2-CH_3_	22.4, CH_3_	1.18, s
3	58.9, CH	4.42, dd (12.0, 4.0)
4	42.8, CH_2_	2.48, dd (14.3, 4.0)
		2.43, d (12.0)
4a	79.2, C	
5	194.0, C	
5a	109.9, C	
6	164.8, C	
7	109.5, CH	6.72, br s
8	164.4, C	
9	108.2, CH	7.17, br s
9a	135.4, C	
10	195.5, C	
10a	83.7, C	
11	41.2, CH_2_	2.68, br d (8.1)
12	115.7, CH	4.70, t (8.5)
13	141.2, C	
13-CH_3_	16.7, CH_3_	1.32, s
14	38.0, CH_2_	2.15, m
		2.10, m
15	33.3, CH_2_	1.93, m
		1.88, m
16	215.0, C	
17	41.1, CH	2.53, m
17-CH_3_	18.4, CH_3_	1.06, t (6.4)
17-CH_3_	18.3, CH_3_	1.06, t (6.4)

**Table 2 molecules-28-00640-t002:** ^1^H (600 MHz) and ^13^C (175 MHz) NMR data of compound **2** in CDCl_3_.

No.	*δ*C	*δ*H (*J* in Hz)
**1**	194.8 s	
**2**	68.0 s	
**3**	67.3 s	
**4**	192.4 s	
**4a**	133.8 s	
**5**	108.3 d	6.89 br s
**6**	164.5 s	
**7**	109.3 d	6.55 br s
**8**	164.8 s	
**8a**	108.7 s	
**1** **′**	25.4 t	3.28 dd (15.5, 7.2)
		2.36 s
**2** **′**	116.9 d	5.11 s
**3** **′**	135.9 s	
**4** **′**	26.0 q	1.73 s
**5** **′**	18.4 q	1.73 s
**1** **″**	25.6 t	2.96 dd (15.5, 7.3)
		2.57 dd (15.5, 5.9)
**2** **″**	120.0 d	5.41 s
**3** **″**	139.3 s	
**4** **″**	77.0 d	4.12 t (6.7)
**5** **″**	33.1 t	2.38 m
**6** **″**	121.7 d	5.30 s
**7** **″**	136.8 s	
**8** **″**	68.8 t	3.98 m
**9** **″**	14.1 q	1.67 s
**10** **″**	12.3 q	1.74 s

**Table 3 molecules-28-00640-t003:** In vitro antiviral activities of compounds 1–7 against PRV in the Marc-145 cells.

Compounds	IC_50_ (μM)	CC_50_ (μM)	TI
**1**	2.056 ± 0.343	30.806 ± 5.565	14.98
**2**	-	-	-
**3**	2.208 ± 0.375	5.790 ± 0.708	2.62
**4**	13.268 ± 1.109	19.630 ± 2.821	1.48
**5**	8.260 ± 0.902	16.680 ± 1.122	2.05
**6**	26.470 ± 5.00	46.318 ± 7.364	1.75
**7**	-	>0.045	-
Ribavirin	58.032	>163.9	>2.82

(1) CC_50_, the cytotoxic effect, the concentration required to reduce Marc-145 cells viability by 50%, was measured using the MTT method. (2) TI, the therapeutic index is defined as the ratio of CC_50_ to IC_50_ (TI = CC_50_/IC_50_). (3) “-” means inactive. (4) Ribavirin was used as positive control.

**Table 4 molecules-28-00640-t004:** Antibacterial activities of compounds **1**–**7** (MIC μg/mL).

Code	*S. aureus*	*S. suis*	*E. rhusiopathiae*	*E. coli*
**1**	25	100	50	-
**2**	-	-	-	-
**3**	<0.78	3.125	50	-
**4**	<0.78	6.25	50	-
**5**	-	25	25	-
**6**	50	50	25	-
**7**	<0.625	20	40	-
Penicillin G	-	12.5	50	-
Cefpirome Sulfate	<0.78	<0.78	<0.78	25

“-” means inactive.

## Data Availability

The data presented are available in the manuscript and Appendix A.

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
