# Peer review of "Napyradiomycin A4 and Its Relate Compounds, a New Anti-PRV Agent and Their Antibacterial Activities, from Streptomyces kebangsaanensis WS-68302"

_molecules, 2023, doi:10.3390/molecules28020640_

Round 1
Reviewer 1 Report
Please see the attachment

Author Response
Dear reviewer#1:
Thank you very much for the comments, we considered and response each of comments point by point. Our response to the comments is listed in the response document, please see the attachment.
Best regards,
Yani Zhang

Reviewer 2 Report
An interesting paper
The structural characterisation by NMR is well done.
The discussion of the mechanism of the compounds is weak and should expended
L15. Full name of PRV
L15-20. Provide errors on IC50 values and other data
L26. “the porcine disease infected by…” This is an incorrect sentence
L35-36. Grammar issue. May be revised to … bacteria such as Streptococcus suis …
L36-37. “S. suis, an important zoonotic”. Missing a verb
L41. …have three main…
L41. Delete including as all forms are listed
L44. It’s not the antibiotics that are resistant.
L47. semi-naphthoquinone nucleus and isoprene unit or a monoterpenoid are included as classes of meroterpenoids, but they are not
L53. mycetes to be demonstrated: incorrect grammar
L60-64. If the compounds were identified in a previous study, why are they being structurally being characterized here?
L154-155. Substituent at position C-16. It is better to specify what subtituent is a c-16 of compound 1 and how that differs with at in at the same position on compound 3.
L157-158. Explain how hydroxyl substitution at C-4¢¢ and C-8¢¢ of 5 are affecting the activity
L194-1937. Compound 2 and other are being referred as know compounds available literature data
L247-261. Why were other fractions not purified?
Author Response
Dear reviewer:
Thank you very much for the comments, we considered each of the comments point by point, and our response to the comments is listed in the response document, please see the attachment.
Best regards,
Yani Zhang
